# Poor School Academic Performance and Benign Epilepsy with Centro-Temporal Spikes

**DOI:** 10.3390/bs13020106

**Published:** 2023-01-28

**Authors:** Luigi Vetri, Annamaria Pepi, Marianna Alesi, Agata Maltese, Lidia Scifo, Michele Roccella, Giuseppe Quatrosi, Maurizio Elia

**Affiliations:** 1Oasi Research Institute-IRCCS, Via Conte Ruggero 73, 94018 Troina, Italy; 2Department of Psychology, Educational Science and Human Movement, University of Palermo, 90144 Palermo, Italy; 3Department of Human Studies-Communication, Education and Psychology, LUMSA University, 00193 Roma, Italy

**Keywords:** poor academic performances, Benign Epilepsy with Centro-Temporal Spikes, rolandic epilepsy, specific learning disorders, learning difficulty

## Abstract

Background: Poor academic performance of students with epilepsy seems to be a multifactorial problem related to difficulties in reading, writing, math, and logic skills. Poor school and academic performances refer to learning problems in a specific academic area due to learning disorders and learning difficulties not excluding the ability to learn in a different manner during school and academic life. Sometimes, school, academic difficulties, and Rolandic epilepsy can coexist together, and there may be comorbidities. Consequently, the risk of impaired academic performance in people with epilepsy is high. Methods: This review analyzed the relationship between Benign Epilepsy with Centro-Temporal Spikes (BECTS) and poor school and academic performance (PSAP) in children and adolescents (aged 6 to 19), and in adults (aged 20 to no age limit). The PRISMA guideline was used to guide our review strategy. Results: This research shows that Benign Epilepsy with Centro-Temporal Spikes (BECTS) and poor school and academic performances are strongly correlated. An early onset age, as well as a long persistence of seizures, correlate more closely with PSAP. On the other hand, it appears that good pharmacological control of seizures and remission from the acute phase of the pathology support better school performance. Conclusions: This review highlights how neuropsychological aspects are also involved in patients with BECTS and PSAP, both in the greater predisposition to the establishment of other neuropsychiatric conditions and in the possibility that stigma conditions and poor academic results may have repercussions on the adaptation and functioning of these subjects. Global management of the subject with BECTS and PSAP is essential, which also pays attention to the aspects of social and scholastic inclusion, both to achieve age-appropriate educational and behavioral objectives, to give the necessary tools for the growth of the individual, and to allow a serene transition to adulthood, favoring autonomous learning and better outcomes.

## 1. Introduction

Specific learning disorders are neurodevelopmental disorders that are typically diagnosed in early school-aged children. The main feature is a persistent impairment in at least one of these major areas: reading, writing, and math [1,2]. Specific learning disorders, in particular, are a set of heterogeneous disorders that may impair the ability to read, write, calculate, listen, and express oneself verbally. They include dyslexia, dysgraphia, dyscalculia, and dysorthography [3]. They are specific, because the disorder only affects some skills, leaving intact the general intellectual functioning; they are also evolutionary, as they vary with the age of the subject. The indicator is a discrepancy between normal quotient general intelligence (IQ ≥ 85)) and skill in a specific domain. Learning disorders are categorized as mild, moderate, and severe [1].

The skills involved are reading, spelling and grammar, and calculation. The lack of these skills causes problems in learning subjects such as Italian, math, science, and geography and may impact everyday activities and social interactions. Students with learning disorders and lower academic achievement can have problems throughout life, owing to the emotional motivational consequences. These problems include psychological distress, depression, unemployment, and dropping out of school [3].

An important note about terminology related to specific learning disorders is a medical term used for diagnosis, which is also abbreviated as “learning disorder”. “Learning disability” is instead a term used by both the educational and legal systems, even if such a term is not exactly synonymous with specific learning disorders [2]. In fact, the term learning disability includes students with cognitive, motor, and sensory problems who are not targeted in this review. Furthermore, there is the learning difficulty that defines a condition causing difficulties in comprehending or processing information, which may be due to several different environmental and social factors. Hence, poor school and academic performances include different definitions and atypical learning trajectories that have different definitions such as learning disability, learning disorder, or learning difficulty [4,5].

Learning difficulty is a special education need that involves areas of learning such as reading, writing, spelling, and mathematics. There are several levels of learning difficulties, for example, one specific learning difficulty is a particular difficulty in learning to read, write, or spell. Learning difficulties do not affect general intelligence (IQ). Some examples of specific learning difficulties are dyspraxia and attention deficit hyperactivity disorder (ADHD) [3,6]. 

In this review, PSAP includes specific learning disorders or learning disorders and learning difficulties. PSAP will be particularly analyzed and related to students with a particular type of epilepsy.

The factors influencing this association are determined by the etiology of epilepsy, the presence or absence of drug resistance and the relative frequency of critical episodes, the type of crisis, and drug therapy. In general, the prevalence of learning disorders appears to be higher in older children and in those in which epilepsy has presented an earlier onset. It is significant that the type of epilepsy (i.e., the type of crisis and etiology) can affect the “stable” aspects of cognitive functions such as school performance, memory, and IQ. Furthermore, paroxysmal epileptic activity, that is, epileptic anomalies and seizures, affect functional cognitive aspects such as speed of reasoning, attention, and short-term memory.

There is a part of the population with epilepsy, in the context of learning disorders, who do not present cognitive delays but may encounter more sectoral difficulties in school learning as in the case of specific learning disorders of reading, writing, calculation, or motor skills. The most significant studies in the literature have been carried out on benign Rolandic epilepsy in childhood, in which difficulties in these specific areas, particularly in writing and reading skills, may be more present than in the general population [7]. 

Young students with epilepsy have to face situations of disadvantage and sometimes even marginalization, especially in school and academic environments. All of this, in addition to jeopardizing self-efficacy and self-esteem, can negatively affect school and academic learning. Despite the development of scientific knowledge, a series of prejudices and clichés revolve around epilepsy, making the life of the person affected even more difficult, along with compromising not only their psychological balance but also their normal integration into social life [4].

Even if there are possibilities for controlling seizures on the clinical level, many problems are not still completely solved for so many patients, such as impaired cognitive functioning, which can globally and permanently alter the development of intelligence or cause specific disabilities. The topic of cognitive disorders in epileptic children is heterogeneous and complex. Epilepsy itself certainly does not determine an intellectual deficit, but disorders of specific neuropsychological functions have been reported in subjects with normal IQ. Both neuropsychological symptoms detected are heterogeneous, as are the epileptic syndromes themselves, and include speech, memory, or attention deficits, visual-spatial deficits, and learning disabilities [7].

Problems concerning cognitive functioning, such as short-term memory deficits or deficits in attention, can be the causes of poor school and academic performance (PSAP) in children with epilepsy [8,9].

Education is fundamental for every individual but children with atypical neurodevelopment trajectories often have academic performance disorders. Students affected by PSAP and Rolandic epilepsy can bear important consequences in scholastic and academic skills. 

### 1.1. Poor School Academic Performances in Students with Benign Epilepsy with Centrotemporal Spikes

People with PSAP have an atypical learning trajectory compared to normality, avoiding any misconception of defining a lack of learning as being cognitively disabled [10].

Children, adolescents, and adults with PSAP have special educational needs. They need, therefore, as students, adequate and personalized support according to their physical, biological, physiological, psychological, and social conditions. There are different categories of pupils with PSAP including those with specific developmental disorders (dyslexia, dysorthography, dyscalculia, dysgraphia) and attention deficit hyperactivity disorder, and students with epilepsy [11,12].

PSAP could be caused by ineffective literacy processes, inadequate teaching methods, ineffective learning styles; disadvantaged economic conditions, and above all a disadvantaged socio-economic context. Furthermore, social factors such as repeated school, home changes, and serious health problems leading to long hospitalizations can contribute to PSAP. Emotional disadvantage is also important, as numerous situations of abuse and mistreatment have been reported in these children [13].

Family problems may compromise the emotional-motivational aspect of the child and therefore learning. It is evident that these conditions are not decisive for a child to have a learning difficulty, but nevertheless, they can influence the learning process. In general, epilepsies do not affect intelligence, learning ability, and performance. However, seizures and drugs can temporarily impair learning ability and performance. Many children with epilepsy do not experience difficulties in school but performance problems can occur [14,15,16,17,18].

In addition, it is also important to remember that learning problems may arise in a stable way during illness or be a temporary condition linked to a particular stage of that child’s illness. It is important to recognize this and to intervene appropriately with specific activities to support learning deficiencies [14]. The most significant studies in the literature were carried out about benign epilepsy with centrotemporal spikes (BECTS) in childhood, in which difficulties in writing and reading skills may be more present [15,16].

### 1.2. A Particular Type of Poor School and Academic Performance: Specific Learning Disorders

In the latest version of DSM-5, specific learning disorders are classified among neurodevelopmental disorders. These disorders have onset during the years of school education and are characterized by persistent and progressive difficulties in learning basic school skills [6,19,20].

The main characteristic of this category is its “specificity”: the disturbance affects a specific and limited domain of essential skills in learning (reading, writing, calculating), without compromising general intellectual functioning. This means that in order to have a diagnosis of a specific learning disorder, the child cannot be affected by intellectual disability or other impairments that could be the main cause of the learning disorder [1,2,3].

Specific learning disorders begin during school age, although they may not be recognized until adolescence or even adulthood. Specific learning disorders refer to ongoing problems in one of three areas, reading, writing, and math [21,22,23].

An estimated 80 percent of those with learning disorders have reading disorders (commonly referred to as dyslexia). Other specific skills that may be impaired include ability in writing, spelling, reading comprehension, mathematical calculation, and mathematical problem-solving. Difficulties with these skills may cause problems in learning subjects such as history, math, science, and social studies, and may impact everyday activities [14].

There are four types of specific learning disorders. Dyslexia is typically manifested by a lack of proficiency in reading. It is about being able to read correctly the sounds and words of your own language. In the case of dyslexia, the ability is called coding of written language in oral language. Dysorthography is typically manifested by a lack of proficiency in writing. It is about being able to spell correctly the sounds and words of your own language. In the case of dysorthography, the ability is named transcoding of oral language into written language. It includes phonological and non-phonological errors. Dysgraphia involves the control concerning the graphic aspects of handwriting as well as the motor-executive aspects of writing. Dysgraphia compromises the quality of the graphic aspect of handwriting to the point of making it illegible and affecting the ability to write fluently. Dyscalculia corresponds to difficulty in learning mathematics. Dyscalculia involves numerical and calculating ability [1,2,3].

Specific learning disorders, if not recognized and followed up, can cause problems throughout a person’s life in addition to lower academic achievement. These problems include an increased risk of higher psychological distress, poorer mental health, unemployment/under-employment, and dropping out of school [1].

A specific learning disorder is a condition that can cause an individual to experience problems in a traditional classroom learning context. It may interfere with developing literacy skills and mathematics, and can also affect memory, focus ability, and organizational skills. A child or adult with a learning difficulty may require additional time to complete assignments at school and can benefit from strategy instruction and classroom accommodations, such as material delivered in special fonts or the ability to use a computer for taking notes [1,2].

Two individuals with a learning difficulty are never exactly alike and many conditions such as dyslexia, attention deficit disorder, attention deficit hyperactive disorder, dyscalculia, and dysgraphia exist on a wide spectrum. Dyspraxia, a motor-skills difficulty, can affect a learner’s ability to write by hand and may impact planning skills. It is not uncommon for learning difficulties to coexist with motor-skills difficulties. For example, dyslexia and dyspraxia, or ADHD and dyspraxia can occur together [10].

### 1.3. A Particular Type of Epilepsy: Epilepsy with Centro-Temporal Spikes

The International League Against Epilepsy (ILAE) defines epilepsy as “a disease characterized by an enduring predisposition to generate epileptic seizures and by the neurobiological, cognitive, psychological, and social consequences of this condition”. Fisher, Robert S., et al. [24].

A practical definition of epilepsy allows an early diagnosis with two unprovoked seizures >24 h apart or one unprovoked seizure and a high probability of further seizures or a diagnosis of an epilepsy syndrome [25].

Idiopathic focal epilepsy (IFE) is the most frequent epilepsy syndrome affecting children and it is characterized by the absence of demonstrable brain lesions, usually rare and brief partial seizures, with otherwise abundant interictal EEG abnormalities, and seizures that cease by the end of adolescence [26].

Benign epilepsy with centrotemporal spikes (BECTS), previously known as Rolandic epilepsy (RE), is the most frequent idiopathic focal epilepsy syndrome and it usually affects children in their early school years. BECTS represents 8–25% of childhood epilepsies with an overall incidence of 10–20:100,000 in children aged 3–15 [27].

BECTS are usually characterized by brief hemifacial seizures that may evolve into a focal to a bilateral tonic-clonic seizure. The EEG traditionally evidences a normal background and a normal sleep architecture with mandatory interictal high amplitude centrotemporal spikes or sharp waves increasing in drowsiness and sleep.

The adjective benign refers to a self-limiting epilepsy with an excellent response to antiepileptic drugs and seizures that usually remits spontaneously at a predictable age.

However, the genetic etiology with complex modes of inheritance, being likely responsible for BECTS, also causes a wide spectrum of correlated disorders such as atypical childhood epilepsy with centrotemporal spikes, status epilepticus of BECTS, epileptic encephalopathy with continuous spike-and-wave during sleep, and Landau–Kleffner syndrome.

Today, it is well-known that academic achievement is lower in children with epilepsy (CWE) of normal intelligence compared to healthy controls [14].

However, despite the existence of several literature data about the academic implications of BECTS, traditionally considered a “benign” form of epilepsy, a lack of updated reviewed data is noticed in this area.

The aim of this review is to verify, through the study of international scientific literature, the relationship between poor school and academic performance in students (children and adolescents ages 6–19, and adults ages > 20) with Benign Epilepsy with Centro-Temporal Spikes (BECTS) and IQ (IQ > 85).

## 2. Materials and Methods

### 2.1. Search Strategies

The PRISMA guideline was used to guide our search strategy, data extraction, and methodology [28,29,30]. A search was conducted on 6 databases: PsycInfo, Scopus, PubMed, APA PsycArticles, and EBSCO.

The search for electronic literature databases was dated January 2022. They analyzed all studies assessing Rolandic epilepsy or Benign Epilepsy with Centro-Temporal Spikes (BECTS) and poor school and academic performance (PSAP) in children and adolescents (aged 6 to 19), and adults (aged 20 to no age limit). Missing papers were requested from the study’s authors by email.

### 2.2. Search Used Form and Area

The search strategy was carried out by combining the following keywords with Boolean operators such as AND, OR, and NOT: (Rolandic OR benign epilepsy OR centrotemporal spikes OR idiopathic epilepsy AND poor school and academic performance [Title/Abstract]) AND (dyslexia OR learning disorders OR learning disabilities OR reading disorder OR dysorthography OR dysgraphia OR dyscalculia [Title/Abstract]) OR poor school and academic performances. Rolandic OR benign epilepsy AND Specific Learning Disorder: dyslexia, dysorthography, dysgraphia, dyscalculia. NOT learning disabilities. All the titles and the abstract were analyzed.

### 2.3. Inclusion and Exclusion Criteria

Published articles meeting the following criteria were included: population: children and adolescents (aged 6 to 19) with BECTS and learning disorders and specific learning disorders and adults (aged 20 to no age limit) with BECTS and learning disorders and specific learning disorders; outcome: a quantitative assessment of at least one form of learning disability (specific), academic difficulty, school performance, reading ability, learning disorder, written language, dyslexia, dysorthography, dyscalculia, reading performance, a diagnosis of BECTS; experimental/quasi-experimental design; international study; journal articles in terms of publication type, in the English language.

Reasons why the articles were excluded: studies which examined specific populations with epilepsy and learning disabilities; qualitative assessment of the variables or assessment of other variables; qualitative study; single case; books, unpublished studies, theses/dissertations; national study and journal; other publication languages.

Twenty-three articles were analyzed and included in the review. A flow diagram following the models of Page et al. [30] is included as Figure 1.

### 2.4. Data Extraction

For each item, papers were independently analyzed by two reviewers to carry out data extraction, and disagreements were discussed with two reviewers. The selected articles were organized for the data extraction according to the following variables: digital object ID, article title, abstract, journal title, journal year, authors, epilepsy, learning disorder, participants, age participants, sex, stage of life, design, country, aim of the study, result. A total of 23 studies met the review inclusion criteria.

## 3. Results

### 3.1. Included Study: Description

Seventeen out of twenty-three studies were cross-sectional studies and six were longitudinal studies (five prospective and one retrospective, respectively). Four studies were conducted in the United States, three in the United Kingdom, one both in the United States and in the United Kingdom, three in Italy, two in Brazil, two in Turkey, two in Sweden, one in Spain, one in Korea, one in the Netherlands, one in Austria, one in Greece, and one in France.

Eighteen out of the twenty-three studies had a control group; the remaining five compared their academic results to normal mean values. Twenty studies considered childhood, thirteen adolescence, and three young adults (Table 1).

### 3.2. Characteristics of PSAP in Children with BECTS

All the studies showed some degree of PSAP in people with BECTS compared to controls or reported norms. Children could have a good long-term outcome of neuropsychological impairment. Some longitudinal studies showed that the performance in neuropsychological testing was comparable to population statistics and controls after the remission of BECTS [11,50]. However, other longitudinal studies showed that LD persisted in patients after remission from seizures and epileptic discharges [12,33].

### 3.3. Factors Associated with PSAP in Children with BECTS

#### 3.3.1. Demographic Psychosocial Factors

Male gender is considered a risk factor for dyslexia [26], but only one study suggested that dyslexia seems to be more frequent in males. On the contrary, dyscalculia has no gender difference [37].

There was a positive correlation between intelligence quotient and reading level in children and adolescents with idiopathic epilepsies [51].

Medical history and handedness are not significantly correlated to LD, and there are no substantial differences in the socioeconomic status or parental education level of patients with BECTS developing LD [36,48]. By contrast, a study showed that educational performances and familial maladjustment are correlated to learning difficulties, attention disorders, and auditory–verbal or visual–spatial deficits [33].

#### 3.3.2. Onset Age

Younger age at epilepsy onset has been identified as a risk factor for reading and math disorders [52]. Moreover, the association between worse neuropsychological functioning and worse reading and writing achievement is stronger in children with early-onset seizures [53].

Several studies have found in BECTS no significant differences regarding the onset age of epilepsy and the presence of any postmorbid neuropsychiatric problems [9,24,27,28,33,34,35,36,37,38,39,40,41,42,43,44,45,46,47,48,49,50,51].

On the contrary, Piccinelli et al., 2008 [40] suggested that seizure onset before the age of eight is a relevant marker for identifying patients at risk of developing academic difficulties, and Oliveira et al., 2010 showed that children having their first seizure at an early age had worse LD compared to those with later onset seizures.

Similarly, a longer epilepsy course seems to be a risk factor for behavioral problems. [54]. Cognitive and academic difficulties appear either shortly after the onset of epilepsy or sometimes before the first seizure [45].

Moreover, the presence of a premorbid neuropsychiatric concern predicts a longer epilepsy duration (*p* = 0.02), a higher seizure count (*p* = 0.02), and a postmorbid psychiatric or neurodevelopmental diagnosis (*p* = 0.002) in patients with BECTS [50].

#### 3.3.3. Seizure Frequency, Duration, and Type

It is well known that seizure frequency and long-duration seizures are correlated to neuropsychological impairment [55].

Conflicting evidence appears in the literature about the relationship between seizure frequency and seizure duration and LD. Some studies indicated that LDs are not significantly correlated to seizure frequency in people with BECTS [11,37].

On the contrary, other studies found a correlation between seizure frequency and duration and some degree of educational performance impairment [33,41,42].

There is no significant correlation between LD and timing of seizure occurrence (during sleep, *p* = 0.768; awake, *p* = 0.942), between LD and type of seizure [43], or between LD and seizure count [50]. Similar results were found in other studies [34,47].

#### 3.3.4. EEG Anomalies

Studies about the effects of interictal epileptiform discharges (IEDs) on cognition have produced mixed results over the years. Most researchers conclude that IEDs, especially when they are highly frequent, can impair cognitive performance in children [56,57].

Centrotemporal EEG discharges in the left or right hemisphere have been associated with LD and language dysfunction [58,59], deficits in oromotor praxis [60,61], executive functions, attention deficits [62], and cognitive and behavioral impairment [63].

However, opposite evidence is also available and finds no correlation between interictal spikes and neuropsychological status [64,65].

A prospective study in patients with BECTS demonstrated that the epileptiform discharges on EEG during sleep were positively correlated to higher attention deficit and higher impulsivity [66].

Piccinelli et al., 2008 [40] evidenced that EEG abnormalities (spikes, sharp waves, spike, and waves) lasting more than 50% of the sleep EEG recording over more than a year are relevant risk factors for developing LD.

Likewise, another study confirmed that children with BECTS and a high spike index during sleep showed worse semantic verbal fluency (*p* = 0.02) and interference condition in the Stroop test (*p* = 0.02) compared to children with BECTS and a low high spike index during sleep [36].

Similarly, Ebus et al. [40] evidenced a significant negative correlation between the amount of nocturnal epileptiform activity and reading sentences (R = −0.525; *p* = 0.008) and verbal IQ (R = −0.51 *p* = 0.08). No correlation was found, instead, between reading performance or verbal IQ and diurnal epileptiform activity.

Massa et al., 2001 [33] have demonstrated that subjects with BECTS are at risk for neuropsychological impairments as long as they have six specific interictal EEG patterns: intermittent slow-wave focus, multiple asynchronous spike-wave foci, long spike-wave clusters, generalized 3-c/s “absence-like” spike-wave discharges, conjunction of interictal paroxysms with negative or positive myoclonia, and abundance of interictal abnormalities during wakefulness and sleep.

However, several studies found no correlation between the number of epileptic discharges or laterality of epileptic discharges or unilateral-bilateral spikes and poor cognitive performances or worse academic performance [11,34,37,42,44,47].

As properly suggested by Ebus et al. [41], the correlation between abundant Rolandic EEG abnormalities and LD does not prove any direct causality but rather points out that the spike burden could reflect the severity of the actual phase of BECTS.

#### 3.3.5. Relation between BECTS Epilepsy and PASP

People with epilepsy tend to have more frequent physical problems, as well as higher rates of psychological difficulties, including anxiety and depression, learning problems of varying types and severity, and consequent behavioral and psychosocial difficulties. Epilepsy is also found relatively frequently in children with neurodevelopmental disorders. It is important for each patient to consider the presence of any comorbidities, to allow for early identification, diagnosis, and appropriate management [4,14]. During the process of acquiring school skills, the child may experience some difficulties common in childhood and adolescence. It is known that when learning to read, the ability to identify words initially occurs through the decoding of the words in the component letters, letter by letter, and in groupings. With advancing school level, the ability to read is based on a lexical repertoire, without the need to use phonological decoding. Children with BCECTS presented a higher percentage of errors in words than the control group, demonstrating that the word recognition process is not yet structured, as expected for the age group. Additionally, with regard to non-words, a higher number of errors than controls are detected, but with a greater reading speed, which could suggest both an impulsive mechanism and confusion between words and the non-words index of a less mature lexical system. Therefore, it can be concluded that children with a new diagnosis of Rolandic epilepsy should be screened for disorders of learning because they are conditions with potentially serious sequelae, which are susceptible to early intervention and almost complete resolution. The average age of BCECTS diagnosis is 7 years, whereas the ideal age for reading intervention is when reading skills are formally taught in school. These children can therefore benefit from a specialist evaluation by psychologists and speech therapists at the time of diagnosis of epilepsy [19,20,67].

#### 3.3.6. Neuropsychological Profile in Patients with BECTS: Features of Learning Performances

Learning can be considered a process of change that depends on the combination of three important factors: neurobiological factors, where we refer to neurological development, the integrity of brain functions, sensory aspects, neuropsychological and brain processing functions; socio-cultural factors, where we refer to influence of the historical-cultural, school and family context; psycho-emotional factors, where we refer to influence of personal factors, personalities, emotional states, learning styles, among others [51].

In subjects with PSAP with BECTS, repercussions are frequent both on the emotional motivational level as highlighted in the analyzed study by Croona, C; Kihlgren, M; Lundberg, S; Eeg-Olofsson, O; Eeg-Olofsson, KE (1999) [31]. This should not come as a surprise, considering, for example, the continuous sense of frustration connected with school difficulties or the fact of being able to achieve only poor results or at least slightly above sufficient despite the high effort related to carrying out daily school activities.

The reading, writing, or arithmetic tasks for pupils with BECTS and PSAP are not simple, and above all are not automatic and carried out with great effort as evidenced in several studies [46,47,49].

In several studies, misunderstanding of the difficulties experienced by the child in the surrounding environment is highlighted. Teachers and parents are often led to attribute insufficient academic results as well as the aversion of the child to school activities to lack of commitment and laziness. In other cases, learning difficulties encountered by children with PSAP and BECTS can also be misinterpreted as an expression of more general intellectual problems [12,34,36,43,61].

The child or young person with PSAP and BECTS may exhibit a variety of negative emotional states characterized by performance anxiety, feelings of insecurity and low self-efficacy, anger, sadness, and low self-esteem. This can generalize from the strictly scholastic sphere to that of interpersonal relationships. When one does not intervene early, recognizing the real nature of the difficulties, the perception of poor self-efficacy of the pupil associated with an impotent style of attribution risks leading to a growing motivational disinvestment in the context of school activities, bringing a depressive neuropsychological profile to the student [35].

The presence of a PSAP and BECTS represents a risk factor with respect to the difficulty of progressively developing psychopathological disorders, above all of an internalizing type such as anxiety and depression [31,68].

For a subject with PSAP and BECTS, if placed in a context that recognizes and accepts the nature of his or her specific difficulties, takes into account the peculiarities of his or her learning style, and recognizes and enhances his or her cognitive potential and resources, his or her psychological well-being and social functioning tend to be analogous to his or her peers. There is evidence that the recognition of a PSAP with BECTS at school age can play a protective role with respect to psychological well-being and social functioning, compared to a late diagnosis. On the other hand, the lack of recognition of the relationship between PSAP and BECTS at school age has negative consequences and is capable of structuring feelings of inadequacy, capable of negatively affecting the quality of interpersonal relationships in adulthood [33,69,70].

#### 3.3.7. Specific Learning Disorders and BECTS: Neuropsychological Profile

Specific learning disorders are conditions characterized by a persistent deficit in the acquisition of basic school tools of reading, writing, and arithmetic skills. They are manifested in children having general cognitive abilities, no sensory or neurological deficits or severe psychopathological disorders, and having been able to take advantage of adequate educational opportunities. BECTS together with the PSAP are neurobiological disorders that can be traced back to anomalies in the functioning of rather specific brain areas or circuits, at the basis of equally specific cognitive processes which are crucial, however, for the adequate functionality of school tools. Their etiology must be sought in a complex interaction between numerous genetic and environmental factors that can act from time to time as risk or protective factors influencing the likelihood that a subject manifests learning difficulties in basic school skills and their extent (multifactorial and probabilistic etiopathogenesis) [21,23,71].

Whether BECTS occurs together with dyslexia and dysorthography, it has specific characteristics. In particular, the learning difficulty of dyslexia concerns deciphering reading (or decoding component of the reading), while dysorthography is about the acquisition of spelling skills; both disorders are very frequently associated and are subtended in many cases by phonological processing disorders (phonological awareness, short-term verbal memory, and RAN) being able to interfere with the automatic learning of the correspondence rules between graphemes and phonemes. It is particularly important to recover the pronunciation of words quickly from their orthographic representation and vice versa [5], associated with the dysgraphia that concerns the graphomotor components of writing and is characterized by the slowness and/or poor quality of the graphic stroke and often by the difficulty in interpreting the written words. At the origin, there may be difficulties affecting various basic neuropsychological functions, such as visuo-spatial functions, fine motor planning and control, and procedural memory [70,72].

Subjects with dyscalculia and BECTS manifest a persistent disturbance in the acquisition of basic artistic skills such as reading and writing numbers and / or understanding the associated quantity and calculation skills; in many cases, dyscalculia is associated with dyslexia and/or dysorthography due to a common defect of phonological processing; more rarely, it occurs in an isolated form being attributable to a deficit in basic numerical cognition. If the relationship between PSAP and BECTS is not recognized promptly and without the appropriate forms of intervention, SLDs can represent a significant risk factor for the psychological well-being and social adaptation of the subject [37,38,39,40].

#### 3.3.8. Antiepileptic Drugs

Some anti-epileptic drugs (AEDs) can provoke potential cognitive and behavioral effects [73] but also enhance cognitive function [74]. Therefore, this variable should be considered.

In the studies analyzed in this review, the AEDs used are the following: valproate acid (VPA), carbamazepine (CBZ), oxcarbazepine (OXC), levetiracetam (LEV), clobazam (CLB), clonazepam, lamotrigine (LTG), Phenytoin; Gabapentin, Topiramate, Felbamate, Phenobarbital, and vigabatrin in monotherapy or polytherapy.

Many studies report the type of AEDs used, although only a few analyze the correlation with neuropsychological impairment.

All evidence appears to agree with the hypothesis that the subjects with BECTS under treatment with AEDs did not significantly differ in neuropsychological features from the patients with BECTS without pharmacological therapy [11,12,31,42,43,44,45,46,47,48,49,50].

Only one study analyzed the impact of each drug on LD in patients with BECTS. Children using CBZ did not have worse performances in reading sentences or words than children who did not take CBZ (*p* = 0.945 for sentences and *p* = 0.731 for words). Likewise, children using VPA did not perform worse than children not taking VPA (*p* = 0.664 for sentences and *p* = 0.710 for words. In addition, the number of AEDs taken is not correlated to reading performance (sentences: R = 0.189, *p* = 0.377; words: R = 0.195, *p* = 0.360) [57].

#### 3.3.9. Comorbidity

The relationship between BECTS and various comorbid conditions such as migraine, cognitive and behavioral issues, psychiatric status, and neurodevelopmental disorders has been well documented in the literature.

Some studies have analyzed if there is a frequent correlation between LD and other conditions in patients with BECTS.

Kirby et al., 2017, in their study evidenced that, regarding language, cognitive, and motor functioning, 48% of patients with BECTS had difficulties in two areas, while 9% had difficulties in all three domains.

Similarly, Perkins et al., 2008 showed in their study a high concordance of motor and cognitive deficits in patients with BECTS.

Some studies evidenced that BECTS is strongly comorbid with reading disability and speech sound disorder (SSD). Interestingly, a history of SSD always precedes a history of reading disability, and siblings of patients with BECTS are at high risk for developing reading disability and SSD, but only if the family member with BECTS has LD [38].

On the contrary, for Currie et al., 2018 neither SSD nor developmental coordination disorder was evident in all children with BECTS who had reading difficulties [49].

Different studies stated that patients with BECTS and LD have an increased prevalence of comorbid behavioral disorders, such as impulsivity, aggressivity, and ADHD [11,33,34,44,47,50].

Finally, some reports have indicated a psychiatric comorbidity in the anxiety–depression spectrum in patients with BECTS and LD [11,44,50].

## 4. Discussion

All cross-sectional studies agree that there is a higher incidence of PSAP in subjects with BECTS when it is compared with control groups or with the norm [11,50]. Some prospective studies appear to state that there is a tendency not to reconfirm learning difficulties in subjects with BECTS when evaluated at long-term follow-up in patients treated and free from epileptic seizures [50]. Nevertheless, such studies show conflicting results, with some studies stating that such educational difficulties tend to linger through time [12].

This review then focuses on the factors associated with PSAP in subjects with BECTS. Among psychosocial demographic factors, it has been observed that male sex is considered a risk factor for dyslexia, but not for dyscalculia. Family problems appear to be significant risk factors for learning disorders [33]. Clinicians following BECTS patients with PSAP have to take into account various demographic and psychosocial factors, such as gender, the age of onset of epileptic symptoms, family situation, and the evaluation of attention disorders or neurosensory deficits.

Focusing on the correlation with the clinical aspects of epilepsy, it appears that there are several factors that can affect the academic outcome of the patient suffering from BECTS. An early onset age, as well as a long persistence of seizures, correlate more closely with PSAP. On the other hand, it appears that good pharmacological control of seizures and remission from the acute phase of the pathology support better school performance. These data suggest that clinical and therapeutic work on seizure control is of great importance not only for the prognostic outcome of the epileptic condition but also for learning and neuropsychological abilities. In fact, this review highlights how neuropsychological aspects are also involved in the patient with BECTS and PSAP, both for the greater predisposition to the establishment of other neuropsychiatric conditions and for the possibility that the stigma conditions and poor academic results may have repercussions on the adaptation and functioning of these subjects. It is essential, therefore, to implement a global management of the subject with BECTS and PSAP, which also pays attention to the aspects of social and scholastic integration, both to achieve age-appropriate educational and behavioral objectives and to give the necessary tools for the individual’s growth and allow a peaceful transition to adulthood, favoring autonomous learning and better outcomes. Managing these subjects must be timely and complete, with a global assessment of the neuropsychological profile, which focuses on different school and learning skills, to allow a rehabilitation treatment aimed at gaining the cognitive profile of the patient and achieving a greater adaptation, in order to prevent any psycho-behavioral sequelae. Early recognition of the disease and its pharmacological treatment appear to be protective factors with regard to long-term sequelae, contrasting with the hypothesis of a negative influence of the use of antiepileptic drugs on cognitive skills. This is demonstrated by the absence of data to support this hypothesis in the articles examined by this review. A 2017 review by Wo et al. [14] addresses the topic of idiopathic epilepsies and learning disabilities, but unlike this, it does not focus on a single form of epilepsy (BECTS), but on epileptic conditions in general.

People with a PSAP have trouble performing specific types of skills or completing tasks if left to figure things out by themselves or if taught in conventional ways. Indeed, in patients with BECTS, school performance is not the only aspect that could be partially compromised, as several studies included in this review underline how there could be significative impairments in the neuropsychological profile and executive functions [31]. Not all studies agree on the long-term evolution of these impairments, with some longitudinal studies reporting improvements in conditions after the remission from epileptic seizures [50] and others reporting the persistence of significant differences with control groups [33].

Individuals with PSAP can face unique challenges that are often pervasive throughout their lifespan. Depending on the type and severity of the disability, interventions and current technologies may be used to help the individual learn strategies that will foster future success. Some interventions can be quite simplistic, while others are intricate and complex. Current technologies may require student training to become effective classroom supports. Teachers, parents, and schools can create plans together with tailored interventions and accommodations, in order to aid the individuals in becoming successful independent learners. A multi-disciplinary team frequently helps to design the intervention and to coordinate the execution of the intervention with teachers and parents. This team frequently includes school psychologists, special educators, speech therapists (pathologists), occupational therapists, psychologists, teachers, literacy coaches, and/or reading specialists.

There is a complex interplay between learning ability and epilepsy which may be the cause of the risk for students with epilepsy of being excluded and marginalized in school. Children with epilepsy should be able to attend school and experience schooling like any other child. The existence of good communication between family and school is significant for the promotion of inclusion. Inclusion is also promoted by the skills of teachers, the awareness of the disease, and the most effective educational strategies for the epileptic student. Furthermore, experts agree that a good education for children with epilepsy is based on the synergy between family and school. Teachers and school staff often have not been adequately trained to support children with epilepsy. Additionally, students with epilepsy are excluded from school activities such as assemblies, sports, after-school clubs, and field trips. A report from the World Health Organization (WHO) Global Campaign has found that epilepsy causes school and social exclusion [75]. Epilepsy experts and educators with expertise in this area agree that students with epilepsy need to be actively involved in all school and extracurricular activities in order to achieve a positive development outcome [52]. Children with epilepsy are often vulnerable, and in this sense, school staff must be aware of the support these children need at all stages of education. It is essential for teachers to learn how to manage an epileptic seizure and then explain in class to everyone what happened to the epileptic partner [76].

To sum up, it is important to involve the child with epilepsy in the teaching-learning process as well as to make them aware that teachers and school staff are able to manage their seizures in the classroom. Obviously, informing teachers and classmates must always be done in full respect of the wishes of the student with epilepsy. Consequently, the child must feel included in the decisions without ever feeling a sense of shame about his/her condition [77].

### Limitations

This review encountered some limitations. First, most of the articles included in this review have a transverse study design and, therefore, multi-center prospective studies should be scheduled in order to verify the validity of the extracted conclusions of this review. In most publications, the researchers did not reach a total sample size of 100 people, limiting the robustness of the studies. Furthermore, in 5 studies out of 23, the sample was not compared to a control group.

## 5. Conclusions

Students with epilepsy face several challenges within the school system including learning, emotional, behavioral, and social adjustment. Learning or behavioral issues specifically associated with epilepsy may not be clearly understood, leading to inappropriate classroom and student management techniques. Epilepsies can be directly responsible for attention and concentration disorders and learning problems. Although epilepsies can exist without obvious clinical manifestations, there are relationships between the level of attention, learning tasks, and epileptic activity [15,28].

Furthermore, tasks that are not interesting enough, frustrating, and boring, can in fact determine an increase in epileptic activity in subjects at risk. Furthermore, school learning requires the functional integrity of the cortical areas responsible for the integration of cognitive processes [78,79].

Even in the absence of lesioned pathologies in epileptic subjects, an epileptic focus can determine the dysfunction of the affected area with consequent learning problems. Furthermore, the most frequently used antiepileptic drugs can interfere with alertness, attention processes, and memory. Therefore, a person with epilepsy during the morning hours may be excessively sleepy if he / she has taken large doses of drugs during breakfast. It can especially bring consequences for school learning due to the slow reaction times of people with epilepsy [17,18].

The consequences of epilepsy for learning problems trigger a spiral process that also involves the behavioral and emotional sphere. Epileptic people are often introverted and/or aggressive, as well as unable to express emotions. Students with epilepsy need to be emotionally supported through specific training courses aimed at enhancing self-esteem and self-efficacy.

The relationship between epileptic children and others is complicated, owing to the consequences of epilepsy [80,81]. The school environment should be supportive of children with epilepsy. While respecting the privacy of the child with epilepsy, the school should support learning issues through a personalized learning plan to promote educational success [14,28]. The school must pay attention to the management as well as the students suffering from epilepsy. It is advisable, in addition to the personalized teaching plan, that the school has a specific internal regulation to govern procedures to be implemented for pupils with epilepsy. Furthermore, it is essential to train school staff that can intervene during an epileptic seizure [4,82]. To sum up, a child with epilepsy can achieve educational success with the synergy of school, family, and environment.

## Figures and Tables

**Figure 1 behavsci-13-00106-f001:**
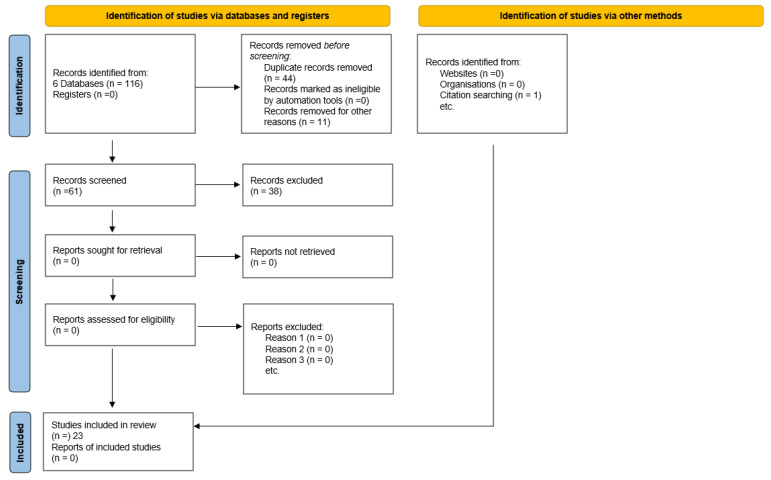
PRISMA flow diagram for literature search.

**Table 1 behavsci-13-00106-t001:** Analyzed studies assessing academic performance disorders in children with BECTS.

Reference	Source	Country	Design	Population	Academic Performance Disorders	Results
Neuropsychological findings in children with benign childhood epilepsy with centrotemporal spikes	Croona et al., 1999 [31]	Sweden	Transverse study	Sample: 17 Age Range: 7–14 Mean age: 12.5 Male/Female: 7/10 Control group: Yes	Learning difficulties	The group with BECTS performs worse than the control group regarding auditory-verbal memory and learning as well as in executive functions
Neuropsychological long-term outcome of rolandic EEG traits	Carlsson et al., 2000 [32]	Sweden	Transverse study	Sample 15 Mean age: 15.9 Male/Female: 10/5 Control group: Yes	Dyslexia	Significant difference between the groups in reading ability of non-related words
EEG criteria predictive of complicated evolution in idiopathic rolandic epilepsy.	Massa et al., 2001 [33]	Italy	Prospective study	Sample 35 Age Range: 12–16 Male/Female: 21/14 Control group: Yes	Learning difficulties	Long-lasting persistence of EEG abnormalities in patients with BECTS is a risk for neuropsychological impairments
Written language skills in children with benign childhood epilepsy with centrotemporal spikes	Papavasiliou et al., 2005 [34]	Greece	Transverse study	Sample 32 Age Range: 7–16 Mean age: 10.1 Male/Female: 17/15 Control group: Yes	Written language skills	BECTS patients performed significantly worse than controls in spelling, reading aloud, and reading comprehension; presented dyslexic-type errors; and had below-average school performance
Epileptic disorders: international epilepsy journal with videotape	Sart et al., 2006 [35]	Turkey	Transverse study	Sample 30 Mean age: 10.8 Male/Female: 21/9 Control group: Yes	neuropsychological and mathematics abilities	Significantly lower performance in drawing, digit span, verbal learning, object assembly, similarities, and vocabulary
Reading abilities and cognitive functions of children with epilepsy: influence of epileptic syndrome	Chaix et al., 2006 [36]	France	Transverse study	Sample 12 Age Range: 7–12 Male/Female: 7/5 Control group: Yes	Reading abilities and learning disorders	The results in the BECTS group did not deviate significantly from the mean of the healthy population
Are dyslexia and dyscalculia associated with Rolandic epilepsy? A short report on ten Italian patients	Canavese et al., 2007 [37]	Italy	Transverse study	Sample 10 Mean age: 10.5 Male/Female: 5/5 Control group: No	dyslexia and dyscalculia	Dyscalculia and dyslexia might be more frequent than expected in children with BECTS
High risk of reading disability and speech sound disorder in rolandic epilepsy families: case-control study	Clarke et al., 2007 [38]	United States	Transverse study	Sample 55 Mean age: 10.0 Male/Female: 39/16 Control group: Yes	reading disorder	BECTS is strongly comorbid with reading disability and speech sound disorder
Benign rolandic epilepsy—perhaps not so benign: use of magnetic source imaging as a predictor of outcome	Perkins et al., 2008 [39]	United States	Transverse study	Sample 9 Mean age: 8.56 Male/Female: 3/6 Control group: No	dyscalculia, and/or expressive language deficits	fine motor dysfunction, visuomotor integration deficits, dyscalculia, and/or expressive language deficits in all of the 9 patients evaluated
Academic performance in children with rolandic epilepsy	Piccinelli et al., 2008 [40]	Italy	Transverse study	Sample 20 Mean age: 10.3 Male/Female: 8/12 Control group: Yes	reading, writing, and calculation disabilities	Specific difficulties with reading, writing, and calculation in 9/20 children with BECTS correlated with increase in epileptiform discharges during sleep and an early onset of epilepsy
Speech and school performance in children with benign partial epilepsy with centro-temporal spikes	Völkl-Kernstock et al., 2009 [11]	Austria	Prospective study	Sample 20 Age Range: 6.0–14.11 Male/Female: 11/9 Control group: Yes	School performance	Worse performance in BECTS group in expressive and receptive grammar
Neuropsychologic impairment in children with rolandic epilepsy	Ay et al., 2009 [12]	Turkey	Prospective study	Sample 35 Age Range: 6–14 Male/Female: 19/16 Control group: Yes	Learning disabilities	Impaired visuomotor, reading ability, and attention to verbal stimuli compared with control subjects. Reading disability persists after remission from seizures and epileptic discharges.
School performance and praxis assessment in children with Rolandic Epilepsy	Oliveira et al., 2010 [13]	Brazil	Transverse study	Sample 19 Age Range: 7–12 Control group: Yes	School performance	Worse performance in BECTS group in writing, arithmetic, and reading
Reading performance in children with rolandic epilepsy correlates with nocturnal epileptiform activity, but not with epileptiform activity while awake.	Ebus et al., 2011 [41]	Netherlands	Retrospective study	Sample: 26 Age Range: 6–12 Male/Female: 13/13 Control group: Yes	Reading performance	Delays in learning efficacy for reading words for reading sentences and negative correlation between amount of nocturnal epileptiform activity and reading sentences and Verbal IQ
Impact of benign childhood epilepsy with centrotemporal spikes (BECTS) on school performance.	Miziara et al., 2012 [42]	Brazil	Transverse study	Sample: 40 Age Range: 7–13 Male/Female: 24/16 Control group: Yes	Academic performance	Children with BECTS showed the worst results in the digit span test, followed by the similarities test
A neurocognitive endophenotype associated with rolandic epilepsy.	Smith et al., 2012 [43]	United Kingdom	Transverse study	Sample: 13 Age Range: 8–16 Mean age: 10.10 Male/Female: 9/4 Control group: Yes	Dyslexia	Evidence of reading impairment in probands but impairments of attention and language were more marked
Cognitive and other neuropsychological profiles in children with newly diagnosed benign rolandic epilepsy	Kwon et al., 2012 [44]	Korea	Transverse study	Sample: 23 Mean age: 9.0 ± 1.6 Male/Female: 13/10 Control group: No	learning difficulties	Evidence of learning difficulties, attention deficits, and aggressive behavior
The neuropsychological and academic substrate of new/recent-onset epilepsies	Jackson et al., 2013 [45]	United States	Transverse study	Sample: 22 Age Range: 8–18 Male/Female: 12/10 Control group: Yes	Academic difficulties	Decreased cognitive performance on measures of academic achievement (arithmetic), language (confrontation naming), verbal learning and memory, and cognitive and psychomotor slowing
Rolandic epilepsy and dyslexia.	Oliveira et al., 2014 [46]	Spain	Transverse study	Sample: 31 Age Range: 7–14 Control group: Yes	Dyslexia	Dyslexia occurred in 19.4% and other difficulties in 74.2% of patients with BECTS
Risk factors for reading disability in families with rolandic epilepsy.	Vega et al., 2015 [47]	United Kingdom–United States	Prospective study	Sample: 108 Age Range: 3.6–22 Male/Female: 66/42 Control group: Yes	Reading disability	Reading disability was reported in 42% of patients with BECTS and it is associated with a history of SSD, ADHD, and male sex but not with seizure or treatments.
Benign childhood epilepsy with centrotemporal spikes (BECTS) and developmental co-ordination disorder	Kirby et al., 2017 [48]	United Kingdom	Transverse study	Sample: 21 Age Range: 8–16 Male/Female: 12/9 Control group: No	School performance	Difficulties in language, cognitive, and motor domains
Reading comprehension difficulties in children with rolandic epilepsy	Currie et al., 2018 [49]	United Kingdom	Transverse study	Sample: 25 Mean age: 9.1 ± 1.7 Male/Female: 16/9 Control group: Yes	Reading disability	Worse performance in reading comprehension and word reading
The natural history of seizures and neuropsychiatric symptoms in childhood epilepsy with centrotemporal spikes	Ross et al., 2020 [50]	United States	Prospective study	Sample: 60 Mean age: 16.0 ± 3.1 Male/Female: 35/25 Control group: Yes	learning difficulties	Learning difficulties common after BECTS diagnosis, but 9-year follow-up, performance on formal neuropsychological testing was normal

BECTS: Benign epilepsy with centrotemporal spikes; SSD: Speech Sound Disorder; ADHD: attention deficit hyperactivity disorder.

## Data Availability

Not applicable.

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
