# Peer review of "Poor School Academic Performance and Benign Epilepsy with Centro-Temporal Spikes"

_behavsci, 2023, doi:10.3390/bs13020106_

Round 1

Reviewer 1 Report

Very well written review, centered on the determination of the existence of any correlation between Benign Epilepsy with Centro-Temporal Spikes and poor school and academic performances. Eventhough the results of this review are well documented, it should be mentioned that multi-center prospective studies should be scheduled in order to verify the validity of the extracted conclusions. This recomendation should be included in the discussion section, as it is not safe to adopt any kind of conclusions based only on review of existing data.

Author Response

Dear Reviewer,

I would like to thank you for your valued comments and suggestions to the article. As you requested, we made all the necessary changes in our manuscript to address your concerns and we detailed how the points raised have been accommodated in the answers to reviewer. The main changes are written in red in the text of the manuscript. From the changes made in the revised manuscript and responses provided below, I hope you are convinced that we have adequately addressed your concerns and made the paper better. If there are any further questions, please feel free to let me know.

Reviewer 1

Very well written review, centered on the determination of the existence of any correlation between Benign Epilepsy with Centro-Temporal Spikes and poor school and academic performances. Eventhough the results of this review are well documented, it should be mentioned that multi-center prospective studies should be scheduled in order to verify the validity of the extracted conclusions. This recomendation should be included in the discussion section, as it is not safe to adopt any kind of conclusions based only on review of existing data.

Thank you for your valued comments and suggestions to our manuscript. As you indicated we added in the “limitation” section of the discussion that “most of the articles included in this review have a transverse study design and, therefore, multi-center prospective studies should be scheduled in order to verify the validity of the extracted conclusions of this review”.

Reviewer 2 Report

Thank you for the opportunity to review the article entitled "Poor school academic performances in Benign Epilepsy with 2 Centro-Temporal Spikes".

This article is a review that aimed to test the correlations between BECTS and academic performance. The review has strengths and merits and focuses on a relevant topic. However, I have some concerns regarding this review.

1) Abstract: I think that the Background in the abstract must be shortened. Moreover, the following sentence "This review highlights how neuropsychological 29 aspects are also involved in the patient with BECTS and PSAP, both for the greater predisposition 30 to the establishment of other neuropsychiatric conditions, and for the possibility that the stigma  conditions and poor academic results may have repercussions on the adaptation and functioning of these subjects" (page 1 lines 29-33) comes in the Results, but I think it is more appropriate to be moved to the Conclusions. 

2) Introduction: this section is repetitive and confusing, it is hard to follow. many phrases were repeated in different places in this section. Additionally, I think that this section is very long and can be shortened. 

3) Materials and Methods: The authors pointed that they excluded cross-sectional design (page 6, lines 283-286), however, they pointed in the description of the included studies that 17 out of 23 studies were cross-sectional studies (page 7 lines 301-302), how can this be explained?

4) Results: this section is clear despite the fact that the authors used small sentences without integration between the results (lines 321-403). 

5) Discussion: this section lacks integration between the results. In addition, there is room to add a limitations section. 

Author Response

Dear Reviewer,

I would like to thank you for your valued comments and suggestions to the article. As you requested, we made all the necessary changes in our manuscript to address your concerns and we detailed how the points raised have been accommodated in the answers to reviewer. The main changes are written in red in the text of the manuscript. From the changes made in the revised manuscript and responses provided below, I hope you are convinced that we have adequately addressed your concerns and made the paper better. If there are any further questions, please feel free to let me know.

Reviewer 2

Thank you for the opportunity to review the article entitled "Poor school academic performances in Benign Epilepsy with 2 Centro-Temporal Spikes".

This article is a review that aimed to test the correlations between BECTS and academic performance. The review has strengths and merits and focuses on a relevant topic. However, I have some concerns regarding this review.

1) Abstract: I think that the Background in the abstract must be shortened. Moreover, the following sentence "This review highlights how neuropsychological 29 aspects are also involved in the patient with BECTS and PSAP, both for the greater predisposition 30 to the establishment of other neuropsychiatric conditions, and for the possibility that the stigma  conditions and poor academic results may have repercussions on the adaptation and functioning of these subjects" (page 1 lines 29-33) comes in the Results, but I think it is more appropriate to be moved to the Conclusions. 

Thank you for this suggestion. As you indicated we shortened the abstract background section and moved the sentence “This review highlights how neuropsychological aspects are also involved in the patient with BECTS and PSAP, both for the greater predisposition to the establishment of other neuropsychiatric conditions, and for the possibility that stigma conditions and poor academic results may have repercussions on the adaptation and functioning of these subjects” in the abstract conclusion section.

2) Introduction: this section is repetitive and confusing, it is hard to follow. many phrases were repeated in different places in this section. Additionally, I think that this section is very long and can be shortened. 

Thank you for this suggestion. We shortened and simplified the introduction as you indicated.

3) Materials and Methods: The authors pointed that they excluded cross-sectional design (page 6, lines 283-286), however, they pointed in the description of the included studies that 17 out of 23 studies were cross-sectional studies (page 7 lines 301-302), how can this be explained?

We sincerely want to thank you for identifying a gross error in our manuscript. Cross-sectional design is not an exclusion criterion of our review and therefore we removed it from the exclusion criteria.

4) Results: this section is clear despite the fact that the authors used small sentences without integration between the results (lines 321-403). 

Thanks for your appreciation. We added new sentences to the discussion to improve the integration between the results of our review. 

5) Discussion: this section lacks integration between the results. In addition, there is room to add a limitations section. 

Thank you for this suggestion. As you indicated we added in the discussion more links to the results of our review. In addition we inserted a limitation section.

Round 2

Reviewer 2 Report

I thank the authors for their revision..

I think that they did a well job and addressed my concerns.. Therefore, I recommend to accept the manuscript